# "On The One Hand…": A Case Study on Designing and Evaluating Controller-Free Gestural Interaction in VR Games

Category: Research

## ABSTRACT

Controller-free interactions allow for virtual reality (VR) experiences to feel more natural and immersive than traditional handheld controllers. However, the process of identifying intuitive controller-free gestures can be complex, as it is difficult to determine the most natural mappings from gesture to game input. This paper highlights a case study from a research collaboration with [company name removed for review process], a game development studio in [country removed for review], on designing and evaluating novel and enhanced gestural interactions for their upcoming commercial VR game [game title removed for review process]. We present lessons learned from this collaboration by introducing a toolset of considerations based on both the players' and game designers' expectations, including recommendations intended to help developers, designers, and researchers in designing and evaluating controller-free gestural controls for immersive experiences.

**Index Terms:** Human-centered computing—Human computer interaction (HCI)——Human-centered computing—Interactive systems and tools

## 1 INTRODUCTION

Understanding the player experience is essential to developing engaging game experiences. The Games User Research (GUR) field is concerned with creating novel and engaging player experiences through iterative design and evaluation, which have become a prominent fixture of the gaming landscape in both industry and academia in recent years. Advances in digital gaming technology and hardware have paved the way for new interactive and unique player experiences [5, 7, 28, 33, 50], including those that use more intuitive and natural input [4, 17, 26]. Many of these experiences include moving away from a more traditional controller scheme and toward one which uses the body as a way to interact. The Nintendo Wii [1] is an early console which made use of player body movement as game input using handheld controllers tracking movement. Later, the Microsoft Kinect [2] used camera-based technology to track the players body in space. However, due to technological and play space limitations, these devices have limited ability to sense the 3D movement of the body in space. On-body platforms allow for more nuanced tracking of signals; Virtual reality (VR) platforms like the Oculus [3] or the HTC Vive [4] have revolutionized how players interact and experience games in a 3D landscape, allowing players to use their full bodies with real-time body tracking. Recent development in controller-free interactions in VR allows users to put away the controllers and use their hand movements as direct input to the game. While controllers rely on mapping buttons to actions, hands are capable of various gestures. Bodily-based interactions have long been researched in the field of HCI, in terms of bodily play experiences [33, 43, 48] and movement-based controllers [28, 41, 49]. However, with each of these various technologies it can be challenging to determine the most natural hand input amongst possible gestures. The process of identifying an effective gesture can be complicated and differs depending on the design context.

In this paper, we present a case study of lessons learned from a collaboration with [company name removed for review process], a game development studio, currently working on a commercial game [title removed for review process] that relies on controller-free gestural interactions (using the Oculus Quest 2) as the primary game input. Through this case study, we prototyped and tested gesture designs for common interactions that occur in this commercial game. We outline a toolset based on the lessons learned for assisting developers with the design and evaluation of novel hand tracking controls for VR which consists of: i) a schema of key theme considerations for assessing gestural interactions ii) a framework for classifying and selecting gestural features for VR. The framework considers three types of interactions: Locomotion, camera rotation, and object interaction. Overall, the toolset is based on the lessons learned through the case study and from research on conceptual frameworks and VR guidelines.

While our reflections are based on the interactions within the commercial game, the results from our case study will help developers and researchers further understand the opportunities and constraints in designing for gesture-based interactions. We hope this case study serves as a launching point to motivate further research on design and evaluation of gesture-only interactions. Furthermore, with games becoming more ubiquitous and embedded into our daily lives [2], this case study contributes to the importance of controller-free gestural interaction design practices especially for development studios (such as our industry research partner) who are eager to differentiate themselves from competitors and increase their commercial value.

## 2 BACKGROUND

The work presented is an interdisciplinary project that utilises insights from different fields. Particularly we looked at the 3D user interaction (3DUI) domain which involves the user interacting in a three dimensional space using either hand or body gestures, or using tracked motion controllers [18]. In addition, our lessons learned contribute to the field of interaction design focusing on the design of interactive experiences based on users' needs [31]. In terms of controller-free hand tracking techniques, there has been several new technologies and input mechanisms that were introduced and studied over the last couple of years such as the Microsoft Kinect [43,48,51], and the Leap Motion [3,5,13,46,50]. These interactions often rely on camera-based motion sensing to read hand or body gestures—which are central to controller-free experiences in virtual and mixed reality games—and provide capability to not use controllers [25]. Similarly, in our work, we focus on the design and evaluation of controller-free interaction for the Oculus Quest 2 VR device, that has built in hand tracking through camera-based input.

In terms of research within a video game context, Rogers [33] prototyped a VR shopkeeper game to study player experience in playful bodily interactions context for varying levels of interaction fidelity. Interaction fidelity has been an important area of VR research, helping explore the level of accuracy the immersive experience can recreate real world interactions to enhance the user experience [27]. Roger's exploration of interaction fidelity leads to a set of guidelines for interaction design in VR games. However, their work and other similar work such as [28, 41, 49], focused on using a controller as an input system. There are also studies that investigated the use of controller-free gestures in VR. For example, Schäfer et al. [40]

---

evaluated gestures for locomotion, exploring two types of variations either a two-hand vs one-hand and index-based vs palm-based interaction. Another example is Ban et al.'s [4] work that looked into determining the target point of pinch gestures. However, these studies did not include longer play duration to get an accurate representation of the impacts on player experience. Another example of locomotion is Wang's [47] work which investigates upslope walking in VR using passive haptic and redirection methods. While they do not use controllers, their system requires adequate space to support continuous movement. For research exploring a particular form of interaction, Farmani et al. [12] evaluated discrete viewpoint techniques for movement and rotation to reduce cybersickness [12]. Other work discusses the design of gestures, including tutorialization of controls [19] and notification/menu placements [34].

Furthermore, one of the interactions we investigate in this paper is climbing. In terms of climbing, there has been work done on virtual limb representation [22], controller-based climbing [7, 23, 24], with the most research done in terms of climbing within mixed or augmented reality environments that combine a physical aid in VR [15, 39, 42, 44]. One distinguishable research is Kosmalla's [21] work that made use of the leap motion to register player grasp input while also incorporating physical props. However, little research exists on controller-free or gesture-based climbing.

Designing gestural controls can be challenging, therefore, in order to design natural and intuitive interactions, we draw inspiration from the concept of a player's mental model, a type of conceptual framework used to understand user needs and requirements, to frame our understanding from both the designer and player's perspectives. Mental model describes the internal mapping a user forms about a system such as what they expect the system to accomplish or how to interact with it; however, a divide between the mental model of the designer and user is a common dilemma in usability [29]. There has been work done in terms of mental model in VR [16, 45, 52], but they lack the applicable insight to understand what common features to account for and specifically addressing the experience of controller-free gestures. In terms of understanding factors to consider when designing gestures, we can look towards work focusing on UX, usability, accessibility [32, 36, 38], VR guidelines and considerable factors [14, 35], natural & intuitive control interaction [17, 26], and 3DUI gesture types [30, 37]. Kim et al. [20] suggested a UX framework for VR with a detailed classification of subparts of the system. Their work inspired us to consider a gesture-based classification framework for VR. VR guidelines for usability and the player experience is also a relevant research area to our work as they describe important factors to consider when it comes to designing immersive experiences. Desurvire et al. worked on the creation of PLAY VR [8], guidelines intended to support game user researchers and designers with usability and playability issues specifically for VR, by adapting them from the PLAY [9] and GAP [10] heuristics. Their work even commented on player perception such as looking towards comfort and usability highlighting an interest to capture player's expectations.

## 3 DESIGN & DEVELOPMENT

The commercial game we used for our research is in development for the Oculus Quest using the Unity 3D game engine, and features novel gestural interactions. The game is a Psychological Horror game in which players explore a sinister, mercurial mountain in search of her missing father.

The main goal of the research team was to assess and recommend improvements to the proposed gestural control scheme within the game. These interactions are summarized below within the context of the three main steps of the interactive design process and outlined in Figure 1. As our focus centered around designing gestural interactions for a commercial game, we adapted a user-centered design (UCD) and Rapid Iterative Testing and Evaluation (RITE)

approach [1, 11]. The research team consisted of seven researchers and developers (4 undergraduate students, 1 Master's student, 1 PhD student, and 1 postdoctoral fellow), and was responsible for the background research, prototyping the alternative gestural interactions, and formative evaluation of the prototypes. The postdoctoral fellow worked closely with the project manager and two designers from the industry partner to oversee project progression and deliverables.

For the prototyping and testing of the gestural controls, we took into consideration the players' needs and requirements (See Fig 3 for a guide of key theme considerations discovered) at each stage of the iterative UCD design and development process. Due to the COVID-19 pandemic, access to external testers and testing locations was limited. Therefore, we primarily relied on expert evaluation done by the research and internal gameplay sessions to assess different prototypes. More information on these can be found in Section 3.3.

### 3.1 Gestural Interactions

Our industry partner implemented an initial set of gestural controls in the game, of which we discuss under broader interaction-based categories below (Locomotion, Rotation, and Object Interactions). The first step the research team took was to document how the interactions are currently implemented, and tested each to note potential usability issues. The second step was to research and develop potential alternatives to help test the efficacy and provide improvements depending on the gesture.

The team performed a competitor analysis by examining gesture controls implemented in other systems such as the Xbox Kinect and MRTK demos for Oculus Quest, and VR games such as The Climb VR [5] and Hand Physics Lab [6]. Below is a list of the interactions implemented in the game, and the naming scheme we will use to describe each within this paper.

**Locomotion**: Movement in the game which includes both upward and forward movement. *Compass Movement* represents the forward and backward movement by Opening the hand, looking at a waypoint, and closing hand to confirm movement to set waypoint. *Climbing* represents the Up and down movement by placing hands on rocks and motioning downwards to navigate upwards on the wall.

**Rotation**: Rotating the field of view left or right. *Thumb Rotation* represents a thumbs-up gesture that's tilted left or right to rotate in the intended direction.

**Object Interactions**: Picking up objects, pressing buttons, and generally interacting with various objects in game. *Pinch and Pull* represents pinching the thumb and forefinger together and moving the hand towards the body regardless of the position of interactable objects. *Physical Touch* represents pressing buttons in the game which requires the player to be in close proximity of the object to interact with it.

### 3.2 Alternative Prototypes

The first phase lead to the design of alternative prototypes and usability modifications to the initial set of gestural controls implemented by our industry partner.

**Locomotion Modifications:** *Waypoint modification* represents a visual feedback adjustment by changing the contrast of the movement waypoints in order to update the visibility of them in certain areas in the game. This modification was recommended to support in accessibility as there were areas where the background colors were similarly colored to the waypoints which affected visibility. *Snap Climbing* represents a snapping

---

[5]https://www.theclimbgame.com/
[6]https://www.oculus.com/experiences/quest/3392175350802835/

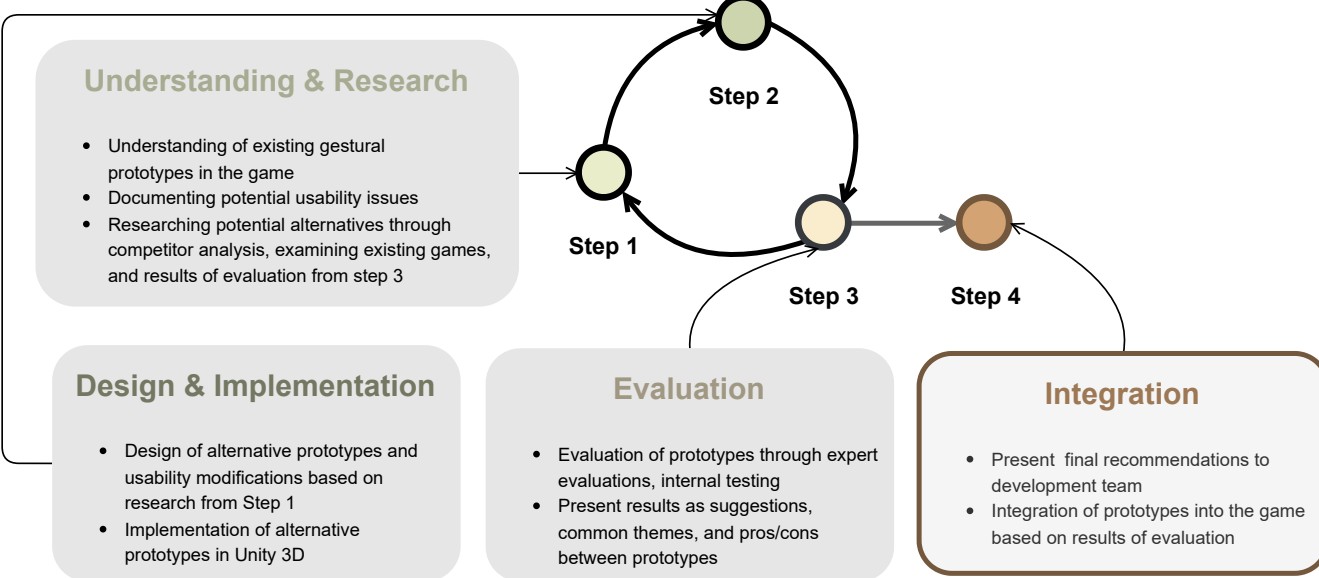

**Understanding & Research**

- Understanding of existing gestural prototypes in the game
- Documenting potential usability issues
- Researching potential alternatives through competitor analysis, examining existing games, and results of evaluation from step 3

Step 2

Step 1

Step 3

Step 4

**Design & Implementation**

- Design of alternative prototypes and usability modifications based on research from Step 1
- Implementation of alternative prototypes in Unity 3D

**Evaluation**

- Evaluation of prototypes through expert evaluations, internal testing
- Present results as suggestions, common themes, and pros/cons between prototypes

**Integration**

- Present final recommendations to development team
- Integration of prototypes into the game based on results of evaluation

Figure 1: Diagram of the Iterative Design and Development Process.

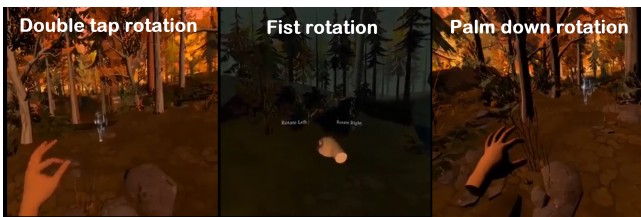

Figure 2: Three of the rotation prototypes developed including *double tap*, *fist*, and *palm down*

feature as a modification to the current climbing implementation that allows the player's hand to snap onto the rock after a grasp motion is performed. *Climbing Outlines* represents a highlight indicator on the rock to know when the player's hand is fully snapped to the rock, and which rocks can be snapped to upon gazing. These modifications were recommended to improve the experience of the climbing gesture by providing more feedback and to provide an intuitive interaction through a snap feature based on our competitor analysis research.

**Rotation Prototypes:** *Fist rotation* represents a fist gesture that is moved sideways left or right in the intended direction. *Palm down rotation* represents an open palm gesture that is oriented downwards and the middle or index finger is tapped together to rotate left or right respectively.[7] *Double tap rotation* represents a double tap gesture using the index and thumb together. Orientation is distinguished using the left or right hand to rotate left or right. See figure 2. These alternative interactions were considered to help brainstorm and assess what would be a natural interaction for gesture-based rotation. We took into account the accessibility of users with limited mobility that might impede them from doing a full 360 rotation and to support players who prefer to play the game seated. These

---

[7]Either hand can be used and the logic is reversed based on the hand that's being used. (eg. index finger+thumb to turn right on left hand, ring finger+thumb to turn right on right hand).

prototypes were utilised for the internal A/B testing that is covered in section 3.3. Stitch Media requested us to assess this as it was an essential component. We put a larger emphasis on understanding this interaction because it is not a frequently implemented interaction in games.

**Object Interactions:** *Sickle Bramble interaction* represents a prototype by making horizontal sweeping motion three times in place of the pinch and pull interaction for a specific action in the game, cutting brambles, using a sickle. Originally only the *Pinch and Pull* interaction was implemented for all object interactions in the game. Based on the literature review highlighting the significance of interaction fidelity for VR games, we sought to evaluate the experience of having high fidelity prototype by proposing the *Sickle Bramble interaction*. *Interactable Object Outlines* represents a modification to interactable objects by providing a yellow highlight indicator on the object to notify the player if they are interactable, and a green highlight is enabled from commencing the interaction (eg. pinch) until the interaction is complete (eg. pull). We proposed this recommendation to help improve the usability of the *pinch and pull* interaction.

### 3.3 Formative Iterative Testing

Based on the proposed prototypes and modifications, the research team and industry partner wanted to evaluate the prototypes and modifications to decide on what interactions and features to integrate into the final version of the game. Given the fast pace of development and short feedback cycles, we decided upon short cycles of iterative testing based on RITE approach [1, 11] consisting of the following: (i) one week to design and develop the prototype, (ii) followed by a week to conduct formative usability test based on expert reviews of gameplay videos from the prototypes, (iii) and a week to prepare a report with our areas-of-improvement notes and suggested recommendations. We found that this process worked effectively for our partnership by providing the research team with adequate time to prepare the report. The format of the playtests consisted of conducting expert evaluations [11] where members of the research team each played the game individually, and took notes

| | Snap Climbing | Climbing Outlines |
|---|---|---|
| **Common Themes** | • Accessibility: allows players to focus on one action at a time rather than having to maintain a hold while also moving their other arm and attempting to grip a new ledge. Helps to make up for the gap which sometimes occurs where players are unable to reach all the way to a handhold.
• Usability: effective feature as it minimizes error and frustration while also being easy to use and satisfying.
• Experience: makes climbing feel smoother without taking away from immersion. | • Feedback: Rocks give clear indication through colour when player's hand is snapped or not
• Visibility: High contract green and yellow outlines help players identify which rocks can be interacted with and if the player successfully grabbed onto a surface |

Table 1: A table describing the common themes discovered from evaluating Locomotion interaction prototypes, *Snap Climb* and *Climbing Outlines*. Common themes emerged when each interaction was evaluated individually.

| | Thumb Rotation | Fist Rotation |
|---|---|---|
| **Advantages** | • Better for smaller jumps in rotation
• More responsive and comfortable
• Less movement required (can feel more natural like moving head slightly or using a controller)
• Easier to control in a more refined state | • Better for larger jumps in rotation
• Can be done in one action (continuous snap rotation) |
| **Disadvantages** | • Tedious to repeat motion
• Must reset thumb up to complete action
• At times game incorrectly recognizes thumb rotation when player attempts palm navigation | • Movement can feel abrupt
• Difficulty making small adjustments to player position or telling distance required to rotate
• Fist can move beyond camera's view |

| | Palm Down Rotation | Double tap Rotation |
|---|---|---|
| **Advantages** | • Can be triggered using only one hand
• Flexible, can be used with either hand to make it more intuitive
• Less tiring | • Intuitive to learn
• Easy to control
• Few accidental triggers |
| **Disadvantages** | • Triggers accidentally because it's only one tap which is similar to pinch and pull. Also triggers a lot in resting position (when hands are placed in the lap)
• Less intuitive initially, needs more tutorialization to orient the player to the controls | • Different orientation of hand changes the feel (double tap feels a lot like pinch & pull vs palm facing upwards to open the compass)
• Might trigger other interactions (like the appearance of the compass if hand is facing up and open) but less often than the palm down rotation |

Table 2: A table describing the advantages and disadvantages from evaluating Rotation interaction prototypes. The advantages and disadvantages where the result of comparing the set of two interactions simultaneously: *Thumb Rotation* and *Fist Rotation*, *Palm Down Rotation* and *Double Tap Rotation*.

| | Sickle Bramble Interaction | Object Outlines |
|---|---|---|
| **Common Themes** | • Immersion: The new interaction better mimics the slicing motion that would be required to use the sickle with a horizontal sweeping motion for further immersion compared to using pinch and pull.
• Interactivity: The new horizontal sweeping motion is more engaging than pinch and pull and diversifies the set of interactions used in the game.
• Intuitiveness: cutting motion is more intuitive than pinch and pull for this situation. | • Feedback: The outline gives a clear indication of whether the player is able to interact with it (outline is yellow) or successfully interacted with (outline is green).
• Visibility: High contract green and yellow outlines help players identify which objects can be interacted with and if the player successfully interacted. The thickness is visible and clear.
• Accessibility: Yellow and Green are two distinguishable colours which are colorblindness friendly. |

Table 3: A table describing the common themes discovered from evaluating object interaction prototypes, *Sickle Bramble Interaction* and *Object Outlines*. Common themes emerged when each interaction was evaluated individually.

on key elements of the experience, and after, the team would discuss the common themes that arose. To support the expert evaluations, we also collected gameplay videos from other players (e.g., from playing different prototypes of thumb rotation vs. fist rotation). Due to COVID-19 restrictions, these players were selected from the immediate connections of the research team members (e.g. roommates, siblings), but still met the target audience of the game and who have not played the game before.

### 3.4 Lessons Learned

We've summarized the lessons learned that emerged from these iterative design and development sprints in table 1, table 2, and table 3 where advantages and disadvantages were discussed when two interaction prototype alternatives were compared against each other, while common themes were discussed when considering the prototype alternative on their own. Table 1 covers the common themes that emerged from evaluating locomotion prototypes. Table 2 covers the advantages and disadvantages of the thumb and fist rotation, and the palm down and double tap rotation interactions. Table 3 covers the common themes in Object interaction prototypes; the two prototype alternatives were assessed individually.

Before the integration phase and as part of the iterative design and development process, two horizontal (full game) playtests were conducted. In the first playtest, 6 players played the game (83.3% were 20 and younger, 16.7% were between the age of 21 to 30, 66.7% have not used VR before, 16.7% have used it once, and the remaining 16.7% have used it more than 10 times) and in the second playtest, 7 players played the game (71.4% were between the age of 21 to 30, and 28.6% were between the age of 51 to 60, 57.1% have used VR 1 to 5 times, 14.3% have used it once, and 28.6% have used it more than 10 times). The horizontal playtest helped us strengthen the lessons learned by assessing the overall usability and user experience of the game along with how the interactions were experienced all together which are detailed in the discussed tables.

### 3.5 Integrate

The research team made recommendations to the industry partner in terms of actionable insights that can be integrated to improve the overall user experience. As the game is still under development, the suggestions that were integrated may differ in the final version of the game.

**Locomotion:** To improve the experience with using compass forward movement, the research team recommended implementing the *waypoint modification* to adjust visibility of waypoints in low contrast areas in the game. Additionally, the team recommended implementing the *snap climbing* feature to enhance the experience, accessibility, and usability with the current climbing implementation and recommended the climbing outlines feature either as an on/off toggle within a menu, or as a permanent feature.

Our goal was to ensure that motion sickness can be decreased, therefore, we also recommended some adjustments to the speed of climbing's upward movement and the player's placement from the wall to ensure that the handholds are within reach. Regarding the *Compass Movement*, a fade-in teleport to the waypoints worked effectively for [removed for review process] as it supported the narrative component to the game by creating areas of interest as well as supported our goal with ensuring motion sickness is minimized.

From locomotion, we noted that it can be expressed as a forward/backward movement (e.g. compass movement) and up/down movement (climbing interaction). In addition, we noted the distinguishing feature between continuous movement or teleportation and their applicability.

**Rotation:** After the internal playtest, most players preferred the *thumb snap rotation* due it being more comfortable, easier to control, and requiring less action to perform. We also noted from the playtest that players were interested in the freedom to choose their desired way of interaction and to be flexible with how they want to interact. Additionally, *double tap rotation* was recommended due to its intuitiveness to learn and simplicity. Thus, the research team recommended making these options available in the game as options which players may select and swap between to suit their needs as a good practice for usability and accessibility giving players the option to choose between one and two-handed options to help account for discomfort players may experience discomfort using both hands for rotation. The fist rotation was dismissed as a possible interaction due to the lack of intuitiveness of the gesture which resulted in players performing the interaction wrong and not achieving the desired outcome (e.g. instead of slight sweep fist motion from side to side, players performed a stronger swing which resulted in discomfort and frustration). Furthermore, we considered the degree of rotation to be another factor to experiment with. We initially recommended that when players rotate their thumb to the left or right, they should only have to do it once for the camera to automatically rotate and include a discrete/continuous toggle in the menus for accessibility. Through internal testing, we discovered it was motion sickness-inducing, so we decided to recommend the discrete motion which rotates in increments.

From Rotation, we noted that it can be expressed as the following: (i) a physical and gesture-based rotation (e.g. *Thumb Rotation*), (ii) it can be performed by one hand or two hands (e.g. *Double Tap Rotation*), and (iii) the rotation can be performed as a continuous rotation or rotate by increments each time the gesture is performed.

**Object Interactions:** The research team suggested implementing the *sickle bramble* feature into the game as a replacement method for cutting brambles with the pinch and pull to enhance the immersion. During the intial playtests and prior to testing the *sickle bramble* interaction, players noted how they would like to be able to interact in a more realistic manner which motivated our implementation of the *sickle bramble* interaction.

From object interaction, we noted that the interaction can be expressed as a direct interaction with the object or as an indirect interaction (e.g. *Pinch and Pull*) where the player and the object is not in close proximity.

## 4 Discussion

Through our iterative design and implementation process, we extend the findings of this case study to explore the design space of controller-free/gestural controls in VR and outline some considerations and recommendations for future designers and researchers working in this space. Here, we present a toolset consisting of common themes that emerged from our design & development process, and a framework to categorize these themes. First, we present our schema: (1) for game designer to understand the main considerations and factors that were considered when designing and evaluating gestures and (2) the player's main expectations from experiencing novel gestures (refer to figure 3). Second, we present a preliminary gestural framework to assist designers with categorizing gestures, and understanding the main differences between different types of gestures to support in selecting which gesture feature is applicable to each game scenario (refer to figure 4). These two contributions comprise our suggested recommendations for gestural controls which we elaborate on in the sections below.

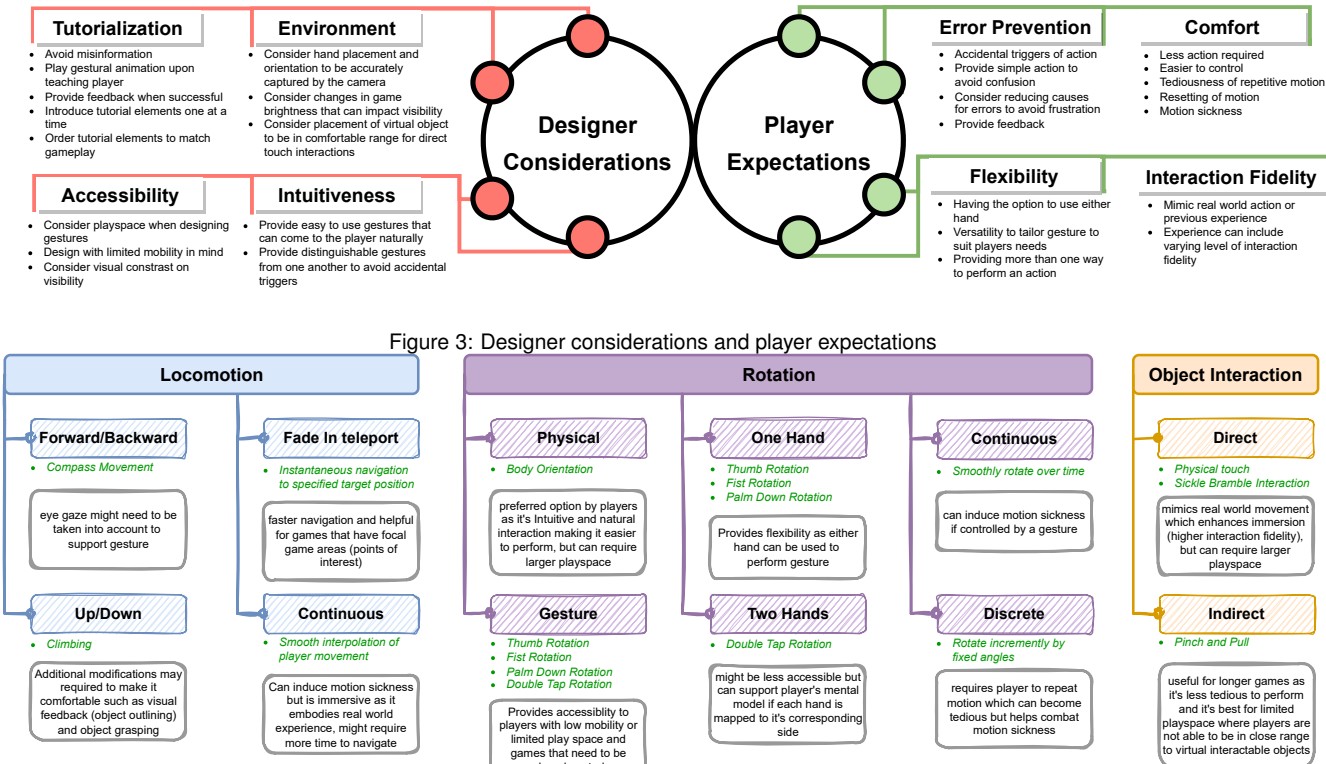

Figure 3: Designer considerations and player expectations

Figure 4: Preliminary Framework of Controller-Free Gestural Controls. An example or description of the subcomponents is provided beneath each feature in green, and followed by the best practices and key takeaways in the text box.

## 4.1 Designer Considerations and Player Expectations for Gestural Controls in VR

In figure 3, we present our insights based on our design and implementation process from this case study. The intended use is for game designers to understand the main considerations and factors to address when designing and evaluating gestures in addition to accounting for the player's main expectations from experiencing novel and enhanced gestures. The main considerations from the designer's perspective was to consider **Intuitiveness**, **Environment**, **Tutorialization**, and **Accessibility** of the gesture design.

In terms of gesture design, **intuitiveness** largely ties into the player's mental model with how they expect the interaction to play out. Tapping fingers together on the hand corresponding to the direction the player wishes to rotate is intuitive and builds upon what the player has learned from other actions and the initial gestural calibration. Given that using a sickle requires a cutting motion, the horizontal sweep is a more intuitive interaction than *pinch and pull* for this situation. **Environment** describes the physical and virtual environment experience for the user. The affordances of the particular system and the technical constraints are important to consider when initially designing the interactions. **Tutorialization** highlights key features to the approach to introducing novel interactions to players and the presentation of the information in an immersive experience. The lack of feedback generated by controller-free interaction can require players with additional time to get accustomed to and requires adequate attention to reliable feedback to the player [25]. **Accessibility** can factor in as play space, handedness, and if the player needs to play the game seated or standing. In the game, the player was given the option to sit while playing, allowing a broader range of users to be able to play. In addition, considering handedness will factor in designing gestures that allow either hand to be used increasing accessibility by eliminating the barrier for certain players

with mobility impairments.

The main expectations towards optimal user experience of players experiencing novel and controller-free gestural controls were **Comfort**, **Interaction Fidelity**, **Flexibility**, and **Error Prevention** of the gesture design. **Comfort** is of utmost importance to consider when designing a gesture. For example, doing repetitive tasks for long duration can lead to discomfort and repetitive motion injuries. Accounting for the ergonomic of the interaction and conducting extensive user testing will help designers decide which gestures are too intensive to use for a long period of time. **Interaction Fidelity** describes the level of realism given to particular interaction designs and is largely influenced by player's mental model of real world interaction and the hardware. For example, There were moments where the Oculus was not registering players' hands during certain rotation versions which influenced player preferences (preferring thumb over fist rotation). We found that having varying levels of interaction fidelity did not impact the immersion of the game such as with incorporating the *sickle bramble interaction* along with the pinch and pull as each effectively worked based on the context. **Flexibility** describes providing versatility with how the player chooses to play the game. For example, having the option to use either hand individually rotation action makes it versatile and allows players to tailor it to suit their needs. **Error Prevention** was an important consideration which played a major role in influencing the user experience. For example, players would get frustrated if the gesture they performed was not captured properly by the system or a different action was triggered instead.

## 4.2 Perspective on Categorization of Gestural Controls in VR

Previous studies have investigated VR guidelines, classifications, and considerable factors from the perspective of HCI and UX for Virtual Reality [14, 20, 35]. Understanding the constraints and appli-

cability of controller-free gestural and its sub-components can assist in understanding the influencing factors on the user experience and provide designers, user researchers, and developers with the insight to make informed decisions when it comes to designing the intended experience for their interaction control. In figure 4, we present the second part to our contribution, our perspective into how VR gestural controls can be classified and categorized based on the interaction type. The intended use is for assisting designers with selecting gestural features that would be applicable to their game scenario. The preliminary framework is segmented into three parts with each part representing a type of interaction: Locomotion, Rotation, and Object Interaction. The Locomotion Category is subdivided into two feature groups: the direction (Forward/Backward, and Up/Down) to differentiate between climbing and walking movements, as well as the form of navigation (Fade in teleport and Continuous). The Rotation Category is subdivided into three feature groups: the rotation type (Physical or Gesture) to differentiate whether a gesture-based rotation is used, the hand input (One hand or Two hands), and the rotation experience (Continuous or Discrete) to specify the implementation of the rotation interaction and how it can impact the experience. Lastly, the Object interaction category contains one set of feature group, the object interaction property (Direct or Indirect), which differentiate between proximity distance of the user and the interactable object.

### 4.3 Limitations and Next Steps

In this case study, we discuss our approach to designing and evaluating gestural interaction for the commercial game, [game title removed for review process]. We acknowledge that our findings are not the result of a formal user study but rather a formative iterative testing process that used user-centered design techniques. In the next phase of this project, we plan to conduct a large-scale user study involving a summative evaluation to assess the preliminary toolset and determine the usability and user experience of the gestural design modifications made to the game.

Regarding the presented preliminary toolset, some of the features/changes proposed are also not specific to hand gestures (e.g. the outlines, snapping, and locomotion waypoint modification). These types of recommendations have already been proposed outside of hand gestures to improve overall interactions in 2D/3D games when using, for example, eye-tracking [6]. However, the case study demonstrates the lessons learned in an industry context in addition to how multiple types of interactions are experienced simultaneously. Understanding how interactions function in diverse contexts can influence the direction and objectives for the gestural interaction design space. With that in mind, our first given toolset can be seen as a foundation for future work to grow and improve upon it. For instance, we are aware that object interaction is a crucial component of developing interaction in VR games and simulations, but this project's scope constraints prevented us from giving it more attention. By taking into account other elements like one-handed and two-handed interaction, further research into controller-free object interaction can aid in expanding the object interaction category inside the framework.

Finally, this case study inspires another path for future work: collaborations between academia and industry. By describing our procedure, we can give other researchers useful advice on how to forge successful industry partnerships.

### 5 CONCLUSION

In this case study, we discuss considerations and recommendations from developing and evaluating VR controller-free/gestural control prototypes, in an ongoing collaboration between a university research and development team and an indie video game industry studio, [removed for review process]. We present a preliminary toolset that consists of a schema and a gestural classification frame-work. The schema is divided into two parts: from the perspective of the designers' considerations and players' expectations of the interactions. From the designer's perspective, intuitiveness, environment, tutorialization, and accessibility are the primary factors to consider. From a player's perspective, comfort, interaction fidelity, flexibility, and error prevention were the most important. In addition, we created a preliminary framework for assessing gestural interactions based on these themes that emerged from the evaluation of novel and enhanced interactions explored through the lens of the player's needs and requirements (based on the proposed schema) and VR guidelines/classification research. The framework categorizes locomotion, rotation, and object interactions into factors that might influence interactions amongst users. In the future, we plan to evaluate these recommendations with users, and run workshops to further understand the design space using the suggested criteria. We hope this work might provide a launching point for designers and researchers working on controller-free and hand tracking controls in VR to understand the various opportunities and limitations for this new design space.

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
