# OpenReview forum: ""On The One Hand...": A Case Study on Designing and Evaluating Controller-Free Gestural Interaction in VR Games"
_graphicsinterface.org/Graphics_Interface/2023/Conference — Submitted to GI 2023_

### Official Review · Reviewer_zFBU · 2023-01-11
**Paper review**

**Rating:** 3
**Confidence:** 4

**Review:**

This paper presents the design and evaluation of gesture-based interaction for a VR Game. Authors conducted this work in collaboration with a game development company. The company had an initial set of gestures, and the researchers conducted an iterative design approach to recommend improvements.

This paper adresses a relevant and timely topic. There is still room to improve current gestural interfaces in VR. It is also interesting that researchers collaborate with a games company, which can have their research transferred to a real game. However, the paper itself has several limitations:

It is difficult to understand what is the key novelty of the paper. For instance, how this work compares to previous work proposing guidelines for VR (which is cited in the related work section, but without detailing the difference with the present work).

While the paper seems to initially focus on the gestural interaction, the paper describes some unrelated aspects during the design and implementation, such as the improvement of the visibility by changing the contrast. I would recommend to focus on the gestural interaction, leaving other aspects of the prototype design aside.

First design phase: The paper is missing many details regarding how the first phase of the design cycle was lead: how many people were involved in each iteration, how much did each iteration last, how many brainstorming sessions were conducted, etc... Also regarding the ideation sessions (if there were any), it would be good to report how they were conducted: was there one person testing the game, etc..

Evaluation: Authors decided to conduct expert reviews based on gameplay videos. I see this choice as having many limitations which may have negatively impacted the quality of these reviews. First, experts are biased, as they have an expert understanding of the interaction, and more particularly if they are the ones that proposed the gestures. Second, it seems difficult to evaluate an immersive interaction by simply looking at a video. How did experts identify usability problems by looking at the video ?

The description of the evaluation is actually quite confusing, as it cites both video-based reviews but also members of the research team playing the game, as well as other players. The description of the evaluation process should be revised to more clearly explain the process. For instance, authors mention that two interaction prototype alternatives were compared with each other: who tested these alternatives ? was it a within or between design ? was the order of the alternatives counterbalanced ?

Overall, this paper seems more appropriate as an industrial case submission, rather than a scientific paper, since the contribution and novelty of the work are unclear. Besides, it is difficult to evaluate the quality of the work conducted, as the paper is missing too much details.

---

### Official Review · Reviewer_qcTT · 2023-01-12
**Paper Review**

**Rating:** 4
**Confidence:** 5

**Review:**

This paper presents a case study on designing and evaluating gestural interaction for a VR game.  The authors examine a new VR game from a game company and perform a user centered design and rapid iterative testing and evaluation on the current version of the game with specific focus on the 3D gestural components and how to make them better and more usable.  The focus was on 3 specific interactions involving locomotion, rotation, and object interaction.  Alternative prototype interactions were developed, and their pros and cons were established.  From this information a preliminary framework for controller-free gestural controls and design guidelines were developed.

Overall, the authors present an interesting paper that explores 3D gestural user interfaces for a specific VR game. The work is certainly relevant, and it is important for researchers to build relationships with VR game companies so the work done in academia can be better utilized in the VR game industry.  The paper was well written and easy to follow, and the references seem appropriate.  Exploring 3D gestural user interfaces is certainly not new but providing guidance on alternative interfaces that could potentially improve the user experience is significant. Although there is interesting work in this paper, there are significant problems.

The main problem with the paper, as noted in the limitations section, is that what is presented is just a case study and thus it is difficult to generalize any of the findings without robust user studies.  As it stands, this paper provides an alternative interface and some initial guidelines based on very little empirical data.  Thus, this paper seems to be just too preliminary at this stage.  I would encourage the authors to continue their work and resubmit the paper once it is more mature. In addition, there is a problem with the references.  In many cases the journal or conference is not included.  These should be included in every reference.

---

### Official Review · Reviewer_kV9o · 2023-01-13
**Rather on the fence about this interesting submission**

**Rating:** 5
**Confidence:** 3

**Review:**

In this communication, the authors present a case study of looking at, iteratively developing, and testing, controller-free gestural interaction in a VR Game. Their manuscript is based on a collaboration with a video game company (redacted for anonymity).

The paper reads quite well, and is easy to understand. It references a large number of past works considering the length of the manuscript. I would say that the work presented in the manuscript is relevant for the conference as it seems to fit the topics that can be presented there, and it is a timely topic considering the growing interest in VR games.

I would first like to highlight that I am not an expert in games or VR for games. My main area of expertise includes using VR but in vastly different contexts so I cannot assess how pertinent the authors’ “background” section is and I rate my expertise with a “3 out of 5”. That being said, I found that the number of references was rather on the “high” side of things. Continuing from this point on my lack of expertise in the field, I would say that the limitations of the study do not highlight enough that this applies surely to games but unlikely outside of this field.

The paper presents a specific case study and is of course very focused on this specific case study. I am quite unsure that the lessons learned from the case study are widely applicable in the sense that subsection 4.2 should have been, in my opinion, much more developed than this. First, in order to clarify what the authors bring with respect to past work in this regard. Indeed, while the authors do mention some past work in this section, it is not entirely clear how their work is different or why it is even needed if some past works have already investigated this issue. This should more clearly appear in the paper. Second, this subsection contains what I would argue is the main interesting contribution of the paper and yet it is a very short subsection with very few insights. I guess another way to put it is that, while reading through the authors' contributions in the introduction, I expected a lot more emphasis would be placed on 4.2, its validity, its use, and its importance with respect to past work, but currently the subsections falls short of doing any of these things.

The paper is missing many details on both the design cycle and the evaluation and this makes it sometimes quite a confusing read. Maybe there is something I misunderstood and I look forward to discussing this with other reviewers in order to see if that is the cas.e

While the figures and tables are all very well done, only a handful of them (mostly figure 3) are appropriately described and used in the main text. While I appreciate the use of helpful and nicely designed figures, I would love to main text to also contain/explain their content in some way, which I thought was lacking in this paper.

Overall, I am slightly on the fence with this submission. While I would love to be enthusiastic about it, because I think we, as a field, should be enthusiastic to collaborate with video game companies and that there are a lot of practical insights that could be revealed by such collaborations and I know how difficult publishing papers resulting from such collaborations should be, I feel like the paper kind of falls short of "selling" its contributions and providing them. While the length of the paper seems somehow appropriate for the contributions made, the level of details provided in the discussion is too shallow for me. I am therefore quite on the fence with this paper. I hope that discussing the submission with other reviewers will help me change my mind about the submission as I really want to be positive about it, but I fear that the paper cannot be accepted without significant revisions of the text to include more detailed contextualization, and insights about the contributions.

---

### Meta-Review · Area_Chair_g9fE · 2023-01-17

**Recommendation:** 4
**Confidence:** 4

**Metareview:**

The paper has received three reviews, with ratings ranging from 3 (clear rejection) to 5 ( Marginally below acceptance threshold).

The reviewers have identified several positive points in this work:
- The paper is well written and easy to follow
- The topic is timely and of relevance for the GI community
- Reviewers appreciated that authors build their work on a relationship with a company
- The topic of using gestural interaction for VR games is interesting, and the outcome of this work could be of interest to improve the user experience in VR games
- The paper cites many previous references

However, reviewers also highlight several limitations to the paper in its current form:
- The paper presents a case study, from which it is difficult to generalize any of the findings
- It is unclear how novel the work presented is, and how this work is different than previous work
- The guidelines are based on very limited data
- The design process and the evaluation descriptions are missing many details
- The work seems too preliminary

Overall, my recommendation falls below the acceptance threshold, as the result of the previous concerns and the average of the reviewers ratings. I believe that authors could either try to improve the current paper by adding more details and trying to reframe the contribution, or else target an industrial track in a conference, which could allow them to promote this collaboration.